# First-in-Human Dose-Escalation Study of the Novel Oral Depsipeptide Class I-Targeting HDAC Inhibitor Bocodepsin (OKI-179) in Patients with Advanced Solid Tumors

**DOI:** 10.3390/cancers16010091

**Published:** 2023-12-23

**Authors:** Anna R. Schreiber, Jodi A. Kagihara, Bradley R. Corr, S. Lindsey Davis, Christopher Lieu, Sunnie S. Kim, Antonio Jimeno, D. Ross Camidge, Jud Williams, Amy M. Heim, Anne Martin, John A. DeMattei, Nisha Holay, Todd A. Triplett, S. Gail Eckhardt, Kevin Litwiler, James Winkler, Anthony D. Piscopio, Jennifer R. Diamond

**Affiliations:** 1Division of Medical Oncology, University of Colorado Anschutz Medical Campus, Aurora, CO 80045, USAross.camidge@cuanschutz.edu (D.R.C.);; 2Division of Medical Oncology, John A. Burns School of Medicine, University of Hawaii, Honolulu, HI 96813, USA; 3OnKure, Inc., Boulder, CO 80301, USAklitwiler@onkure.com (K.L.);; 4Livestrong Cancer Institutes, Department of Oncology, Dell Medical School, The University of Texas at Austin, Austin, TX 78712, USA; 5Department of Immunotherapeutics and Biotechnology, School of Pharmacy, Texas Tech University Health Sciences Center, Abilene, TX 79601, USA; 6Dan L Duncan Comprehensive Cancer Center, Baylor College of Medicine, Houston, TX 77054, USA

**Keywords:** HDAC inhibitors, solid tumors, phase I clinical trial, MEK inhibitors

## Abstract

**Simple Summary:**

Histone deacetylase (HDAC) inhibitors are anti-cancer agents that have demonstrated efficacy in hematologic malignancies; however, the utility of available agents is limited, owing to narrow therapeutic indices and suboptimal isoform selectivity. Bocodepsin (OKI-179) is a novel, orally bioavailable, Class I-targeting depsipeptide HDAC inhibitor with promising anti-cancer activity in preclinical solid tumor models. In this first-in-human phase I clinical study, we report the safety and tolerability of OKI-179 administered orally daily with intermittent and continuous dosing schedules. OKI-179 was well tolerated with low rates of high-grade adverse events, supporting the potential for the successful combination of OKI-179 with other targeted anti-cancer agents. OKI-179 is currently being investigated in combination with the MEK inhibitor binimetinib in patients with NRAS-mutated melanoma.

**Abstract:**

(1) Background: Histone deacetylases (HDACs) play a critical role in epigenetic signaling in cancer; however, available HDAC inhibitors have limited therapeutic windows and suboptimal pharmacokinetics (PK). This first-in-human phase I dose escalation study evaluated the safety, PK, pharmacodynamics (PDx), and efficacy of the oral Class I-targeting HDAC inhibitor bocodepsin (OKI-179). (2) Patients and Methods: Patients (*n* = 34) with advanced solid tumors were treated with OKI-179 orally once daily in three schedules: 4 days on 3 days off (4:3), 5 days on 2 days off (5:2), or continuous in 21-day cycles until disease progression or unacceptable toxicity. Single-patient escalation cohorts followed a standard 3 + 3 design. (3) Results: The mean duration of treatment was 81.2 (range 11–447) days. The most frequent adverse events in all patients were nausea (70.6%), fatigue (47.1%), and thrombocytopenia (41.2%). The maximum tolerated dose (MTD) of OKI-179 was 450 mg with 4:3 and 200 mg with continuous dosing. Dose-limiting toxicities included decreased platelet count and nausea. Prolonged disease control was observed, including two patients with platinum-resistant ovarian cancer. Systemic exposure to the active metabolite exceeded the preclinical efficacy threshold at doses lower than the MTD and was temporally associated with increased histone acetylation in circulating T cells. (4) Conclusions: OKI-179 has a manageable safety profile at the recommended phase 2 dose (RP2D) of 300 mg daily on a 4:3 schedule with prophylactic oral antiemetics. OKI-179 is currently being investigated with the MEK inhibitor binimetinib in patients with NRAS-mutated melanoma in the phase 2 Nautilus trial.

## 1. Introduction

Histone modification represents an important regulatory mechanism of gene expression, whereby acetylation leads to increased gene transcription and deacetylation is associated with gene repression [1,2,3]. Histone acetylation is controlled by the opposing activities of histone acetyltransferases (HATs) and histone deacetylases (HDACs), which enzymatically add or remove acetyl groups from histones and non-histone proteins, respectively, thereby modulating chromatin structure [1,2,4]. Global dysregulation of histone acetylation has been linked to oncogenesis and cancer progression in solid and hematologic malignancies [1,4,5]. Increased HDAC expression in cancer leads to the repression of tumor suppressor genes, resulting in increased tumor cell proliferation, decreased apoptosis, loss of cell adhesion, tumor cell invasion, and angiogenesis [1,2,3,4,5].

Eighteen human HDACs are classified into four categories based on homology with yeast deacetylases, cellular function, location, and substrates [3]. Class I HDACs (1, 2, 3, and 8) play a critical role in cell cycle progression, proliferation, and DNA damage response, and have been validated as promising anti-cancer targets in solid and hematologic cancers [6,7,8]. HDAC inhibitors were initially discovered from natural sources and have since been synthetically developed with improved activity and specificity [2,3].

Three HDAC inhibitors have been FDA-approved for the treatment of hematologic malignancies [3,9,10,11]. Vorinostat and belinostat are broad HDAC inhibitors with activity against Class I, II, and IV HDACs [9,10]. Romidepsin is far more selective and potent for Class I HDACs; however, it requires intravenous administration [11]. The pitfalls of the approved HDAC inhibitors include poor isoform selectivity and non-oral delivery routes. The successful development of HDAC inhibitors for solid tumors has been limited by toxicity associated with pan-HDAC inhibitors, insufficient potency of many agents, and suboptimal duration of feasible target coverage with intravenous agents [12]. Therefore, there remains a clinical need for potent, selective, and orally bioavailable HDAC inhibitors.

Bocodepsin (OKI-179) is a prodrug that is metabolized to a largazole-derivative OKI-006. OKI-006 is a potent inhibitor of Class I HDACs (1, 2, 3, and 8; cell-free IC_50_ 1.2, 2.4, 2, and 47 nM, respectively) with limited activity against Class IIa HDACs (IC_50_ > 1000 nM) [12]. In preclinical studies, OKI-179 demonstrated significant antitumor activity against murine (syngeneic) and human cancer cell line xenograft models, including colorectal cancer, triple-negative breast cancer, and diffuse large B-cell lymphoma when administered as a single agent; it significantly enhanced the activity of other agents while demonstrating acceptable tolerability [12,13].

The principal objective of this phase I first-in-human dose escalation study was to characterize the safety profile and determine the maximum tolerated dose (MTD), recommended phase 2 dose (RP2D), and schedule of OKI-179 administered orally with continuous and intermittent dosing in patients with advanced solid tumors. The secondary objectives included the assessment of the pharmacokinetics (PK) of OKI-179 and OKI-006, and pharmacodynamic (PDx) effects on histone acetylation and preliminary antitumor activity.

## 2. Materials and Methods

### 2.1. Study Design

This was a single-center, open-label dose escalation study of OKI-179. This study was open from April 2019 to December 2021. The primary objectives of this study were to determine the MTD, RP2D, and safety of the drug. The secondary objectives were to assess PK, PDx, and efficacy. 

### 2.2. Treatment Regimen and Safety Assessment

OKI-179 was administered orally in three different dosing schedules in patients with advanced solid tumors. In the intermittent dosing cohort, patients received OKI-179 once daily on days 1–4 (4:3) or 1–5 (5:2) every 7 days, repeated for three weeks on a 21-day cycle. In the continuous dosing cohort, patients received OKI-179 once daily continuously without interruption. Patients fasted prior to dosing, and the starting dose was 30 mg/day, calculated at one-sixth of the dose that was the highest non-severely toxic dose in dogs. 

Dose escalation was performed using an accelerated titration design, in which single evaluable patients were enrolled sequentially into escalating dosing cohorts until a grade 2 treatment-related treatment-emergent adverse event (TEAE) was observed, or the cohorts were expanded for additional safety, and PK and PDx data. Subsequent dosing cohorts were enrolled using a standard 3 + 3 design. 

Dose-limiting toxicity (DLT) was defined as any of the following AEs that were at least possibly related to OKI-179 and occurred during cycle 1: grade 4 neutropenia > 5 days, grade 4 thrombocytopenia, any grade thrombocytopenia with hemorrhage, febrile neutropenia requiring antibiotics, any grade ≥3 non-hematological AE (nausea, vomiting, or diarrhea only if >3 days despite supportive care), <75% of planned dosing in cycle 1 due to any AE, delay of starting cycle 2 for >2 weeks due to any AE, or any treatment-related death. If any patient in any cohort of three patients developed a DLT during cycle 1, an additional three patients were recruited to expand the cohort to up to six evaluable patients. If two or more patients in any expanded cohort developed a DLT, this dose was considered non-tolerated, and dose escalation was discontinued. The MTD was defined as the highest dose level for which no more than one of six patients experienced a DLT.

Patients were continuously evaluated for adverse events throughout the study using the Common Terminology Criteria of Adverse Events (CTCAE v. 5). The patients underwent a clinical history and physical examination, an ECOG PS, a complete blood count (CBCs), chemistry, coagulation studies, urinalysis, a serum pregnancy test (as applicable), a triplicate ECG, an ophthalmology examination, and baseline tumor assessment at screening. During cycle 1, weekly history and physical examination, CBC, chemistry, coagulation parameters, urinalysis, and triplicate ECGs were performed; in subsequent cycles, weekly labs were limited to a CBC. Repeat ophthalmology examinations were performed at cycle 3, day 1, and at the end of treatment. Repeat tumor assessment was performed every 2 cycles (6 weeks) in accordance with RECIST 1.1. 

Intra-patient dose escalation was allowed, and the patients were treated until disease progression, unacceptable toxicity, or withdrawal of consent. The protocol was approved by the institutional review board, and written informed consent was obtained from all patients prior to performing study-related procedures in accordance with federal and institutional guidelines (Clinicaltrials.gov identifier: NCT03931681).

### 2.3. Eligibility Criteria

This study enrolled patients ≥18 years old with advanced or metastatic solid tumors that were refractory or ineligible for standard therapy, with an evaluable or measurable disease according to the Response Evaluation Criteria in Solid Tumors (RECIST) version 1.1, an Eastern Cooperative Oncology Group performance status (ECOG PS) of 0–1, and adequate hematopoietic, hepatic, and kidney function. The key exclusion criteria included women who were pregnant or nursing; had radiation or chemotherapy within 3 weeks, had major surgery or monoclonal antibody administration within 4 weeks; treatment with an investigational agent not expected to be cleared by Cycle 1 Day 1; ongoing adverse events related to prior therapies not resolved to Grade ≤ 1 except alopecia or peripheral neuropathy; prior HDAC, pan-deacetylase, heat shock protein 90 inhibitors, or valproic acid for the treatment of cancer; known symptomatic active central nervous system metastasis or carcinomatous meningitis; concomitant malignancies or previous malignancies within 2-years; known HIV- or AIDS-related or active hepatitis B or C; and congenital long QT syndrome or taking medications associated with significant QT prolongation.

### 2.4. Pharmacokinetic Sampling and Assay

Blood samples were collected immediately prior to the first dose on cycle 1 and day 1, and then at 0.25, 0.5, 1, 2, 4, 8, and 24 h after taking OKI-179 on days 1 and 15. Additional samples were collected pre-dose and 2 and 4 h after OKI-179 administration on day 18 in later cohorts. Blood samples were stabilized with sodium fluoride and formic acid immediately after collection to arrest the additional conversion of the prodrug OKI-179 to OKI-006. Plasma was further stabilized using ascorbic acid prior to low-temperature storage.

Plasma concentrations of OKI-179 and OKI-006 were analyzed using validated liquid chromatography-mass spectrometry (LC-MS/MS) (Quintara Discovery Inc., Hayward, CA, USA). OKI-006 contains a free thiol moiety that can reversibly react with endogenous molecules in the blood and tissue, resulting in “bound” OKI-006. The unreacted OKI-006 is considered “free”. As bound OKI-006 can reversibly convert to free OKI-006 in vivo, both the free and “total” (free plus bound) concentrations of OKI-006 are considered pharmacologically relevant and were therefore quantified as part of this study. Bioanalytically, the total OKI-006 was measured by chemically converting all bound OKI-006 to free OKI-006 prior to analysis using the free OKI-006 bioanalytical method. 

Standard non-compartmental PK parameters (e.g., C_max_, time of maximum plasma concentration, T_max_, AUC, and t_1/2_) were estimated for each patient on days 1 and 15. The AUC_last_ was calculated from time zero to the time of the last quantifiable concentration for OKI-179, free OKI-006, and total OKI-006. 

### 2.5. Pharmacodynamic Analysis

Blood samples for acetylated (Ac)-H3K27, Ac-H3K9, and Ac-lysine levels in peripheral blood mononuclear cells (PBMCs) were collected in cycle 1. Samples were collected immediately prior to dosing and at 2, 8, and 24 h post-dosing on days 1 and 15. Blood samples were processed within 3 h of collection, and isolated PBMCs were washed and resuspended in cell cell-freezing medium. Cells were stored at ≤−70 °C prior to shipment to the University of Texas at Austin Dell Medical School (UTDMS) for analysis. Briefly, patient PBMCs were thawed at 37 °C for 5 min, washed with flow wash buffer containing PBS/2% FBS/5 mM EDTA, and stained for 30 min on ice with the following: fixable viability dye (BioLegend 423106; Zombie NIR^TM^, San Diego, CA, USA), CD3 (Biolegend; Clone OKT3), CD4 (BioLegend; Clone SK3), and CD8 (Biolegend; Clone RPA-T8). All samples were washed twice before being fixed and permeabilized using a BD Biosciences (Hunt Valley, MD, USA) Cytofix/Cytoperm™ kit (RRID: AB_2869008) according to the manufacturer’s protocol and stained for 30 min for intracellular acetylation markers: Ac-H3K27 (Cell Signaling Technology; Clone D5E4), Ac-H3K9 (Cell Signaling Technology, Danvers, MA, USA; Clone CFB11), and total Ac-lysine (BioLegend; Clone 15G10). All cells were analyzed on a Cytek Aurora (5 L 16UV-16V-14B-10YG-8R). Spectral unmixing was performed based on fluorophore-specific single-color cells and compensation bead (ThermoFisher 01-2222-41, Waltham, MA, USA) controls using SpectroFlo© software (v3). All post-acquisition analyses were performed using FlowJo software (v10.3). Results are shown as fold-changes from pre-dose values for each patient and day.

### 2.6. Statistical Methods

Descriptive statistics were used for baseline characteristics, safety assessments, pharmacokinetic variables, exploratory assessments, and efficacy analysis, including the duration of response and time to progression. Tables were used to summarize the data across the cohort of patients and various dosing groups. Geometric means were used to calculate plasma concentration-time profiles. Fold changes from baseline were used to calculate acetylation. 

## 3. Results

### 3.1. Patient Characteristics

Thirty-four patients were enrolled between May 2019 and September 2021 at the University of Colorado Cancer Center. The relevant baseline characteristics are presented in Table 1. The median age was 62 years (range 38–83) and the majority of the patients were female (76.5%). Most patients were Caucasian (79.4%), followed by Hispanics or Latinos (11.8%), and African American (8.8%). The most common tumor types included hormone receptor (HR)-positive, human epidermal growth factor receptor 2 (HER2)-negative breast cancer, pancreatic cancer, and ovarian cancer (all *n* = 5, 14.7%). Approximately three-quarters of the patients (73.5%) had received four or more prior lines of systemic therapy. Patients discontinued treatment owing to disease progression (*n* = 24), withdrawal of consent *(n* = 6), discretion of the investigator (*n* = 3), and adverse events (*n* = 1). The patients were on treatment for a mean of 81.2 days (range 11–447, median 39.5 days).

### 3.2. Dose Escalation and DLTs

Doses from 30 to 450 mg were evaluated in the intermittent dosing 4:3 cohort. The dose escalation scheme and DLTs in cycle 1 by dosing cohort are summarized in Table 2. Two patients were enrolled in the 60 mg cohort due to a patient error in dosing. Although a grade > 2 treatment-related TEAE was not observed, the cohorts taking 270 mg or more were expanded to three patients to obtain more safety, and PK and PDx data. At 450 mg in the intermittent 4:3 dosing cohort, one patient experienced a DLT of grade 2 platelet count decrease, resulting in <75% of dosing in cycle 1. The patient’s thrombocytopenia improved with dose-holding and dose reduction. The cohort was expanded to six evaluable patients, and no further DLTs were observed. However, due to the overall toxicity profile, 450 mg with 4:3 intermittent dosing was deemed the MTD and was the maximum administered dose on the 4:3 schedule.

The final cohort consisted of six patients treated with 300 mg of intermittent 5:2 dosing. One patient in this cohort developed a DLT of grade 3 nausea lasting for >3 days that improved with holding the drug to grade 2, but upon resuming treatment at 200 mg, the patient experienced grade 3 anorexia and came off the study. None of the patients in this cohort experienced DLT. 

### 3.3. Safety

All patients experienced a TEAE, and 33 (97.1%) experienced a treatment-related TEAE. Grade > 3 TEAEs were observed in 18 (52.9%) patients. Serious adverse events (SAEs) occurred in 14 (41.2%) patients, one of whom was treatment-related (grade 3 nausea). The most common TEAEs were nausea (70.6%), fatigue (47.1%), decreased platelet count (41.2%), anemia (35.3%), and anorexia (32.4%). A total of nine patients experienced 15 grade > 3 treatment-related TEAEs, including anemia (11.8%), decreased platelet count (5.9%), prolonged QT interval on ECG (5.9%), and anorexia (5.9%). Eleven patients experienced treatment-related TEAEs, leading to dose holding or reduction. 

The most common TEAEs observed in the 4:3 intermittent dosing schedule were nausea (63%), anemia (40%), fatigue (40%), anorexia (35%), decreased platelet count (30%), vomiting (30%), and diarrhea (25%) (Table 3). Grade 4 or 5 TEAEs were not observed on the 4:3 schedule. The most common grade 3 TEAEs were anemia (25%), fatigue (15%), and prolonged QT (15%). No grade > 3 thrombocytopenia was observed in the 4:3 schedule. At the highest 4:3 dose of 450 mg, patients most commonly experienced grade 1 or 2 nausea (83%). 

In the 4:3 dosing cohort, two patients receiving 450 mg required dose reductions to 350 mg due to grade 2 thrombocytopenia (DLT) and grade 2 anemia (non-DLT). Patients in the 60 mg, 180 mg, and 210 mg cohorts were all dose-escalated once the higher doses cleared the safety analysis (Figure 1). In the 180 mg cohort, following a second dose escalation to 350 mg, one patient had to be dose reduced back to 210 mg due to weight loss and prolonged QT.

In the continuous dosing cohort (Appendix A), the most common TEAEs were constipation (75%), nausea (75%), fatigue (50%), platelet count decrease (50%), diarrhea (38%), and headache (38%). Two grade 4 (25%) thrombocytopenia events occurred in the 300 mg cohort, and the dose was reduced to 200 mg in both patients. No other grade 4 events were observed during this dosing cohort. Two grade 3 TEAEs were observed: grade 3 fatigue and grade 3 anorexia. 

In the 5:2 cohort (Appendix A), the most common TEAEs were nausea (83%), fatigue (67%), decreased platelet count (67%), and vomiting (50%). No grade 4 or 5 TEAEs were observed in the 5:2 schedule. One patient was dosed and reduced to 200 mg due to a DLT from grade 3 nausea. 

Three patients overall had a TEAE leading to death, including multiple organ dysfunction, hepatic failure, and hypercalcemia. All of these events were related to disease progression and were not related to OKI-179.

### 3.4. Pharmacokinetic Studies

Measurable concentrations of OKI-179, free OKI-006, and total OKI-006 were detected in all the patients treated with OKI-179. For each analyte, exposures based on AUC_last_ and C_max_ tended to increase with increasing doses over the full dose range (Figure 2A). Exposures after repeat dosing were equivalent to cycle day 1 exposures with no evidence of accumulation. The maximum concentration was typically reached approximately 1-h post-dose, regardless of the dose, for all analytes. 

The geometric mean concentration-time profiles on days 1 and 15 based on the 300 mg 5:2 regimen cohort are shown in Figure 2B. This dose was selected for analysis as it was the closest to the RP2D. The 300 mg mean single-dose C_max_ and AUC_last_ values for OKI-179 were 142 ng/mL and 253 h·ng/mL, respectively (Appendix A). Exposure to free OKI-006 was greater than that of the prodrug (1770 ng/mL and 2790 h·ng/mL) but less than that of total OKI-006 (1990 ng/mL and 6040 h·ng/mL). After repeated dosing, the relative exposures of free and total OKI-006 compared to the prodrug OKI-179 were approximately 10× and 20× higher, respectively. Therefore, approximately half of the circulating OKI-006 is in the bound state at any given time. The terminal half-life (t_1/2_) values of OKI-179, free OKI-006, and total OKI-006 were approximately 1.7, 3.4, and 6.5 h by day 15, respectively. 

### 3.5. Pharmacodynamic Analysis

Using data from days 1 and 15 at all 300 mg doses (this dose was selected as it was relevant to RP2D), histone acetylation in patient cells time-matched to PK collection was found to be similar in CD4, CD8, and total T cells (Figure 3A). We observed a 6-fold increase in histone acetylation 2 h after dosing, which coincided with C_max_ (Figure 3A). Fold changes in histone acetylation in CD4+, CD8+, and total T cells were similar on days 1 and 15. The total OKI-006 concentration-dependent increase in H3K27 acetylation in CD4 cells is shown in Figure 3B. More pronounced fold changes in acetylation tended to occur at higher plasma concentrations, typically at 2–4 h post-dose. Furthermore, H3K9, H3K27, and lysine acetylation fold changes were independent of the day of assessment (see Appendix A for the mean changes aggregated across days). OKI-179 treatment at doses >200 mg resulted in a greater than 6-fold histone increase in H3K27 acetylation in CD4 T cells (Appendix A).

### 3.6. Antitumor Activity

The best response by RECIST 1.1 was SD in 12 of 31 (39%) patients who underwent repeat imaging. No partial or complete response was observed. The mean therapy was 81.2 days. Six patients remained in the study with stable disease for > 5 months, including two patients with platinum-resistant serous ovarian cancer who were in the study for 7.5 and 14.7 months, respectively (Figure 1).

## 4. Discussion

In this first-in-human phase I dose-escalation trial, OKI-179 was evaluated in intermittent and continuous dosing schedules and was generally well tolerated in patients with advanced solid tumors. The most common adverse events in all dosing cohorts were nausea and anemia, while thrombocytopenia was a DLT with continuous dosing. The recommended schedule for OKI-179 administration in future clinical trials is 4:3 intermittent dosing based on overall better tolerability compared to the 5:2 schedule which was associated with more cumulative nausea requiring multiple anti-emetics, and compared to continuous dosing which was associated with dose-limiting thrombocytopenia. The PK profile of the active metabolite, OKI-006, demonstrated linear pharmacokinetics with exposures exceeding the efficacy thresholds observed in preclinical models at doses below the MTD. OKI-179 is a next-generation HDAC inhibitor based on its more selective HDAC inhibition profile and oral administration which allows a more consistent target coverage over multiple days while maintaining a favorable safety profile. OKI-179 is novel compared to approved HDAC inhibitors as it has narrow Class I targeted activity and is administered orally. This unique profile makes OKI-179 promising as a single agent for select hematologic cancers and for use in solid tumors in combination with other targeted therapies, including MEK inhibitors based on a large body of existing literature [14,15,16,17,18].

OKI-179 has a manageable safety profile and common adverse events across all dosing schedules included nausea, anemia, fatigue, decreased appetite, thrombocytopenia, and vomiting. These AEs are common treatment-related AEs in single-agent trials of other HDAC inhibitors but, overall, were low grade for most patients treated with OKI-179 [19,20,21,22]. Thrombocytopenia was an on-target DLT that was observed at higher doses and especially in the continuous dosing schedule. Thrombocytopenia has been observed with other HDAC inhibitors and is likely related to transcriptional repression of hematopoietic factors, leading to a delay in megakaryocyte maturation [23,24]. Patients who developed thrombocytopenia showed rapid improvement in their platelet counts (generally within a week) with the holding of OKI-179, and no clinically significant bleeding was observed. An intermittent 4:3 dosing schedule was selected for future OKI-179 trials to avoid significant thrombocytopenia associated with continuous dosing. 

Nausea was more prevalent in our study at higher doses and in the continuous dosing cohort. Only one grade 3 nausea event was observed, which was a DLT in the 300 mg 5:2 cohort. Despite nausea being grade 1 or grade 2 in most patients, the majority required supportive care with antiemetics to manage nausea. Granisetron and prochlorperazine are the preferred antiemetics for managing nausea related to OKI-179 owing to their limited impact on QTc, and ondansetron was a prohibited medication in this trial. Chronic nausea requiring treatment with multiple anti-emetics was more common in patients treated on the 5:2 schedule and in patients treated on the 4:3 schedule at 450 mg. Overall, patients treated at the lower doses of 270 mg and 350 mg on the 4:3 schedule tolerated OKI-179 better with regard to chronic nausea requiring ongoing management with anti-emetics. The RP2D dose of OKI-179 is 300 mg on the 4:3 schedule. This dose was selected to reduce chronic nausea and prevent significant thrombocytopenia while maintaining sufficient pharmacologic activity based on PK/PDx modeling. Given the interest in using OKI-179 in combination with other targeted therapies, the RP2D was selected to maximize the safety profile.

Due to toxicities experienced in the continuous and 5:2 dosing schedule, and to improve tolerability in future combination trials of OKI-179, the dose of 300 mg in the 4:3 dosing schedule was selected as the RP2D. The most common grade 3 AEs noted in the 4:3 dosing cohort included anemia, fatigue, and QT prolongation. Other HDAC inhibitors have been associated with QT prolongation [25]. In our study, three patients experienced a grade 3 QT prolongation and four patients had grade 1 or 2 QT prolongation. No significant events occurred because of QT prolongation, but patients taking OKI-179 should routinely have ECG monitoring, and QT-prolonging medications should be avoided. No grade 4 AEs were noted in the 4:3 dosing schedule and no grade 5 AEs occurred in any of the cohorts. Overall, OKI-179 had toxicities that were consistent with other HDAC inhibitors.

The PK profile of OKI-179’s active metabolite, OKI-006, demonstrated increasing exposure to increasing doses. With OKI-179 doses of 200 mg or greater, the C_max_ and AUC exceeded the preclinical threshold for efficacy [12]. OKI-179 is a unique depsipeptide HDAC inhibitor with a similar HDAC inhibition profile to romidepsin, however, the terminal t_1/2_ of total OKI-006 on day 15 was 6.5 h, longer than that of romidepsin. Compared to romidepsin, OKI-179 can be administered daily for several days in a row, increasing target coverage, which may be important in efficacy.

OKI-179 doses of 200 mg or greater also resulted in a > 6-fold increase in H3K27 acetylation in circulating T cells. T cells were selected for peripheral evaluation of histone acetylation based on the emerging body of literature supporting immune modulation as a potential mechanism of action for HDAC inhibitors in cancer [26,27,28]. In preclinical models of triple-negative breast cancer and B-cell lymphoma, OKI-179 was found to potentiate the activity of immune checkpoint inhibitors [12]. The changes in T cell acetylation as a response to OKI-179 dosing suggest that combining OKI-179 with immunotherapy could prove efficacious in overcoming immunotherapy resistance and is an avenue that could be further explored.

In a heavily pretreated patient population, 39% of the patients had stable disease as the best response, and the mean time on treatment was 81.2 days. Two patients with platinum-resistant ovarian cancer had prolonged stable disease lasting 7.5 and 14.7 months, respectively. Studies suggest that high-grade serous ovarian cancers have a higher expression of Class I, II, and III HDACs and that the overexpression of these HDACs is correlated with platinum resistance, possibly explaining this result [29,30]. Other patients with prolonged stable disease had adenoid cystic carcinoma, ER+/HER2− breast cancer, non-small cell lung cancer, and Ewing’s sarcoma. Future biomarker work is needed to identify a patient population that is more sensitive to OKI-179. 

In this first-in-human study of single-agent OKI-179 in patients with solid tumor malignancies, the best overall response was SD. However, many preclinical studies have demonstrated that when HDAC inhibitors are used in combination with mitogen-activated protein kinase (MAPK) inhibitors in RAS-mutated cancers, cell death occurs [17,18,31,32]. Maertens et al. found that the combination of the Class I HDAC inhibitor entinostat, with MAPK pathway inhibitors, led to tumor regression mediated by HDAC3 inhibition [17]. Preclinical models evaluating the combination of OKI-179 and the MEK inhibitor binimetinib in RAS-pathway cancers found that the combination is synergistic and results in increased DNA damage, leading to apoptosis [33]. This combination is currently being evaluated in the ongoing Nautilus trial in patients with RAS pathway-mutated advanced solid tumors and NRAS-mutated melanomas (NCT05340621), with phase 2 data demonstrating there was a 38% overall response rate (ORR) in patients with metastatic NRAS-mutated melanoma previously treated with immunotherapy [34]. The encouraging results of this combination suggest that OKI-179 could potentially have clinical efficacy in other RAS-mutated cancers or possibly with other pathway inhibitors.

## 5. Conclusions

OKI-179 is a novel, oral, Class I targeting HDAC inhibitor that displayed acceptable toxicity, PK, and PDx in this first-in-human study. OKI-179 was overall well tolerated with a low incidence of high-grade adverse events. Prolonged stable disease was observed in a subset of patients, including platinum-resistant ovarian cancer. This study supports the continued investigation of OKI-179 in combination with other targeted therapies, including MAPK pathway inhibitors, in solid tumors.

## Figures and Tables

**Figure 1 cancers-16-00091-f001:**
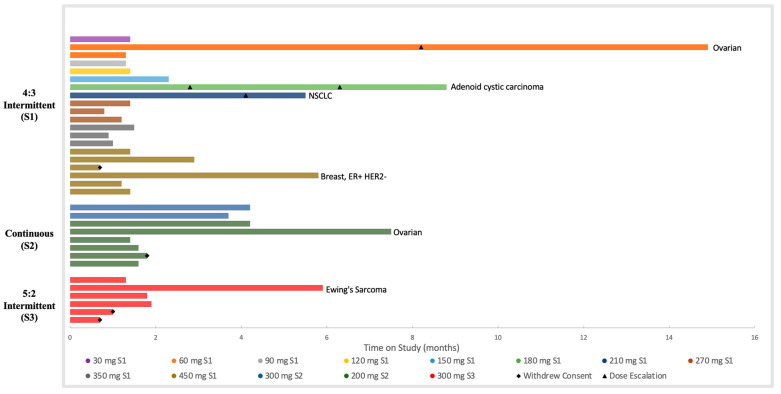
Duration on OKI-179 treatment. Each bar represents an individual patient, and each color corresponds to a dose level. Schedule 1 (S1): intermittent dosing 4 days on 3 days off. Schedule 2 (S2): continuous dosing. Schedule 3 (S3): intermittent dosing 5 days on 2 days off. Durable responders (on study ≥5.5 months) included patients with ovarian cancer, adenoid cystic carcinoma, non-small cell lung cancer, HR+/HER2− breast cancer, and Ewing’s Sarcoma.

**Figure 2 cancers-16-00091-f002:**
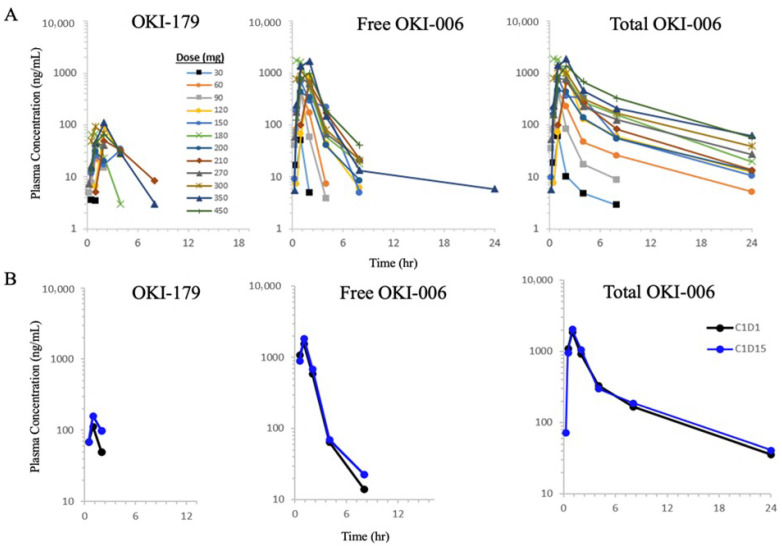
Plasma concentration-time curves. (**A**) OKI-179, free OKI-006, and total OKI-006 on Cycle 1 Day 1 for all dosing cohorts, and (**B**) Cycle 1 Day 1 and Cycle 1 Day 15 for the 300 mg 5:2 Cohort.

**Figure 3 cancers-16-00091-f003:**
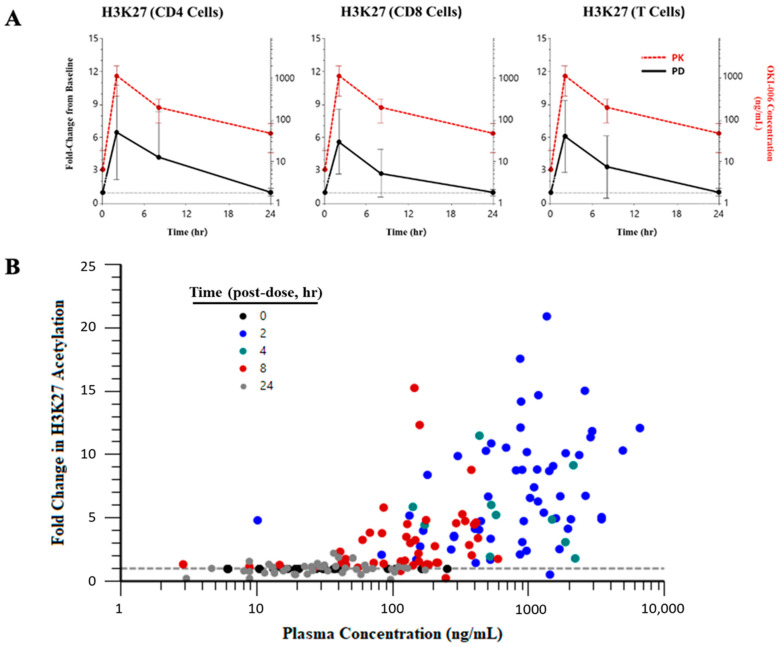
Change in histone acetylation in peripheral blood circulating T cells with OKI-179 treatment and total OKI-006 plasma concentration. (**A**) Fold change from baseline in H3K27 acetylation (black line) following a single dose of OKI-179 300 mg on Cycle 1 Day 1 or Cycle 1 D15 in CD4+ T cells, CD8+ T cells, and all T cells. Plotted with OKI-006 concentration over time (red line). (**B**) Fold change from baseline in H3K27 acetylation in CD4+ T cells versus total OKI-006 plasma concentrations for all patients.

**Table 1 cancers-16-00091-t001:** Baseline Demographics and Patient Characteristics.

Characteristic	Number of Patients (%) *n* = 34
Age, years	
Median (range)	61 (38–82)
Gender	
Female	26 (76.5%)
Male	8 (23.5%)
Race/ethnicity	
Caucasian	27 (79.4%)
Hispanic or Latino	4 (11.8%)
African American	3 (8.8%)
Tumor type	
Breast (ER+ HER2−)	5 (14.7%)
Pancreatic	5 (14.7%)
Ovarian	5 (14.7%)
NSCLC	4 (11.8%)
Colorectal	2 (5.9%)
Endometrial	2 (5.9%)
Melanoma	2 (5.9%)
Appendiceal adenocarcinoma	2 (5.9%)
Other ^#^	7 (20.6%)
Baseline ECOG performance status	
0	16 (47.1%)
1	18 (52.9%)
Prior lines of systemic therapy	
1	2 (5.9%)
2	3 (8.8%)
3	4 (11.8%)
>3	25 (73.5%)

Abbreviations: ER, estrogen receptor; HER2, human epidermal growth factor receptor 2; NSCLS, non-small cell lung cancer; ECOG, Eastern Cooperative Oncology Group. ^#^ Other tumor types included one patient each with triple-negative breast cancer, cholangiocarcinoma, adenoid cystic carcinoma, prostate cancer, cervical cancer, Ewing’s sarcoma, and laryngeal cancer.

**Table 2 cancers-16-00091-t002:** Number of Patients with DLTs per Dose Level.

Cohort	Dose	Patients	DLTs
Intermittent Dosing (Schedule 1)			
1	30 mg	1	0
2	60 mg	2 ^§^	0
3	90 mg	1	0
4	120 mg	1	0
5	150 mg	1	0
6	180 mg	1	0
7	210 mg	1	0
8	270 mg	3 ^†^	0
9	350 mg	3 ^†^	0
10	450 mg	6	1 ^z^
Continuous Dosing (Schedule 2)			
11	300 mg	2	2 *
12	200 mg	6 ^#^	0
Intermittent Dosing (Schedule 3)			
13	300 mg	6	1 ^a^

Abbreviations: DLT, Dose-limiting toxicity; PK pharmacokinetic; Schedule 1, intermittent dosing with 4 days on 3 days off; Schedule 2, continuous dosing; Schedule 3, intermittent dosing with 5 days on 2 days off. ^§^ Two patients enrolled because one patient was not evaluable due to a dosing error. ^†^ Cohort was expanded in order to obtain more safety and PK data not due to AEs. ^z^ <75% of dosing in cycle 1 administered due to grade 2 thrombocytopenia. * <75% of dosing in cycle 1 administered due to grade 2 thrombocytopenia and grade 4 thrombocytopenia ^#^ One patient in cohort 11 moved to cohort 12 after a single dose of 300 mg due to 2 DLTs in cohort 11. ^a^ Grade 3 nausea. Based on the favorable tolerability in the 4:3 cohort, continuous dosing was initiated starting with 300 mg daily. In this cohort, the first two patients experienced DLTs (*n* = 1, grade 2 platelet count decreased, resulting in <75% of dosing in cycle 1 and *n* = 1, grade 4 platelet count decreased). In both patients, the platelet count returned to baseline with a brief dose hold, followed by a dose reduction to 200 mg. A third patient was enrolled in the 300 mg continuous dosing cohort but was reduced to 200 mg after cycle 1 day 2 due to the two DLTs in the cohort. For analysis purposes, this patient was included in the 200 mg continuous dosing cohort. Six patients in total were treated in the 200 mg cohort, and no DLTs were observed, making 200 mg daily the MTD for continuous dosing.

**Table 3 cancers-16-00091-t003:** Treatment-Emergent Adverse Events in ≥15% of Patients Treated with Intermittent Dosing of OKI-179 of 4 Days on and 3 Days Off (4:3).

Adverse Event *n* (%)	30 mg–210 mg (*n* = 8)			270 mg (*n* = 3)			350 mg (*n* = 3)			450 mg (*n* = 6)			All 4:3 Patients (*n* = 20)			
	G1	G2	G3	G1	G2	G3	G1	G2	G3	G1	G2	G3	G1	G2	G3	All Grades
Nausea	5 (63)	0	0	1 (33)	0	0	1 (33)	1 (33)	0	2 (33)	3 (50)	0	9 (45)	4 (20)	0	13 (63)
Anemia	0	2 (25)	2 (25)	0	0	1 (33)	1 (33)	0	0	0	0	2 (33)	1 (5)	2 (10)	5 (25)	8 (40)
Fatigue	2 (25)	1 (13)	1 (13)	1 (33)	1 (33)	0	0	0	1 (33)	0	0	1 (17)	3 (15)	2 (10)	3 (15)	8 (40)
Decreased appetite	3 (38)	1 (13)	0	0	0	0	0	1 (33)	0	1 (17)	1 (17)	0	4 (20)	3 (15)	0	7 (35)
Platelet count decreased	0	0	0	1 (33)	0	0	2 (67)	0	0	1 (17)	2 (33)	0	4 (20)	2 (10)	0	6 (30)
Vomiting	4 (50)	0	0	0	0	0	0	1 (33)	0	1 (17)	0	0	5 (25)	1 (5)	0	6 (30)
Diarrhea	5 (63)	0	0	0	0	0	0	0	0	0	0	0	5 (25)	0	0	5 (25)
ECG QT prolonged	0	0	1 (13)	0	0	0	0	0	1 (33)	1 (17)	0	1 (17)	1 (5)	0	3 (15)	4 (20)
Muscular weakness	0	1 (13)	0	1 (33)	0	0	0	1 (33)	0	0	0	1 (17)	1 (5)	2 (10)	1 (5)	4 (20)
Back pain	1 (13)	0	0	0	0	0	0	0	1 (33)	2 (33)	0	0	3 (15)	0	1 (5)	4 (20)
Pyrexia	2 (25)	0	0	0	0	1 (33)	0	0	0	1 (17)	0	0	3 (15)	0	1 (5)	4 (20)
Cough	2 (25)	1 (13)	0	0	0	0	0	0	0	1 (17)	0	0	3 (15)	1 (5)	0	4 (20)
Proteinuria	0	0	0	0	0	0	0	0	0	0	3 (50)	0	0	3 (15)	0	3 (15)
Hypokalaemia	1 (13)	0	0	0	0	0	0	1 (33)	0	0	1 (17)	0	1 (5)	2 (10)	0	3 (15)
Weight decreased	1 (13)	1 (13)	0	0	0	0	0	0	0	1 (17)	0	0	2 (10)	1 (5)	0	3 (15)
Dyspnea	2 (25)	0	0	0	0	0	1 (33)	0	0	0	0	0	3 (15)	0	0	3 (15)
Headache	2 (25)	0	0	0	0	0	0	0	0	1 (17)	0	0	3 (15)	0	0	3 (15)
Nasal congestion	1 (13)	0	0	0	0	0	0	0	0	2 (33)	0	0	3 (15)	0	0	3 (15)

Graded using National Cancer Institute Common Terminology Criteria for Adverse Events (CTCAE), version 5. Abbreviations: Gr, Grade; *n*, Number. There were no grade 4 or grade 5 TEAEs in the patients treated with the 4:3 schedule.

## Data Availability

Data is available on request due to restrictions. The data presented in this study are available on request from the corresponding author. The data are not publicly available due to patient privacy.

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
