# Peer review of "First-in-Human Dose-Escalation Study of the Novel Oral Depsipeptide Class I-Targeting HDAC Inhibitor Bocodepsin (OKI-179) in Patients with Advanced Solid Tumors"

_cancers, 2023, doi:10.3390/cancers16010091_

Round 1

Reviewer 1 Report

Comments and Suggestions for Authors

The manuscript is suitable for publication.  Please check the few comments in the following to improve your manuscript: 

1. Author needs to explain the loop fall of the other three HDAC inhibitors that have been FDA-approved.

2. Why did the author not include one TNBC patient with another breast cancer patient.  3.  Did the author find any difference in dose-escalation, OKI-179 in different cancer types and different subtypes of cancer such as breast cancer ? 4. Author needs to clarify the novelty of this study in comparison with already approved HDAC inhibitors. 5. What specific improvements could the authors consider regarding the

methodology?

Author Response

  1. Author needs to explain the loop fall of the other three HDAC inhibitors that have been FDA-approved.

Comments were added in the introduction addressing the issues with the three approved HDAC inhibitors.

  1. Why did the author not include one TNBC patient with another breast cancer patient.

Since there were 5 patients with HR+ breast cancer patients and only one patient with TNBC, we thought it was best to separate these based on sub-type.

  1. Did the author find any difference in dose-escalation, OKI-179 in different cancer types and different subtypes of cancer such as breast cancer ?

This was not specifically looked at. Since the sample size was so small, it would be difficult to assign perceived differences as due to a difference in cancer type.

  1. Author needs to clarify the novelty of this study in comparison with already approved HDAC inhibitors.

A sentence was added to the conclusion paragraph. OKI-179 is novel in that is has narrow class I targeted activity and is delivered orally (instead of IV like romidepsin).

  1. What specific improvements could the authors consider regarding the methodology?

Treating more patients in expansion cohorts to further assess efficacy would have facilitated a better understanding of single agent activity.

Reviewer 2 Report

Comments and Suggestions for Authors

This manuscript details the phase I clinical study conducted to test the efficacy of Bocodepsin (OKI-179) an orally bioavailable, Class I-targeting HDAC inhibitor. Few minor comments:

1. The authors could mention in the abstract that this trial is a dose escalation study in the abstract for better clarity.

2. The current ‘study design’ contains information more pertinent to ‘Treatment regimen and safety assessment’ – which the authors could consider renaming this subsection. For the study design aside from retaining line 95-96, the study design section could include primary and secondary objectives of the phase I trial, exploratory objectives if any, as well as the patient recruitment period.

3. The statistical methods should be more detailed on what statistical methods were used for the data analyses performed.  

4. Did patients who had received a particular or multiple line of systemic treatment respond better to OKI-179 than others (maybe like a priming effect)?

Author Response

  1. The authors could mention in the abstract that this trial is a dose escalation study in the abstract for better clarity.

This was added to the abstract.

  1. The current ‘study design’ contains information more pertinent to ‘Treatment regimen and safety assessment’ – which the authors could consider renaming this subsection. For the study design aside from retaining line 95-96, the study design section could include primary and secondary objectives of the phase I trial, exploratory objectives if any, as well as the patient recruitment period.

More information was added to “study design” including primary, secondary objective and patient recruitment period. A new section called “treatment regimen and safety assessment” was made per the reviewers suggestions.

  1. The statistical methods should be more detailed on what statistical methods were used for the data analyses performed.  

Further information on statistical methods was added.

  1. Did patients who had received a particular or multiple line of systemic treatment respond better to OKI-179 than others (maybe like a priming effect)?

There was no obvious correlation to response and prior lines of treatment.